# Limited and idiosyncratic thermal acclimation in soil saprotrophic fungi

Maria Moreno-Druet[1,2*], Anna Maria Kondrat[1], François Rineau[1], Nadejda A. Soudzilovskaia[1], Frederik De Laender[2]

1 Environmental Biology Research Group, Centre for Environmental Sciences, Hasselt University, Diepenbeek, Belgium, 2 Research Unit in Environmental and Evolutionary Biology (URBE), Institute of Life-Earth-Environment (ILEE), Namur Institute for Complex Systems (NAXYS), University of Namur, Namur, Belgium

* maria.morenodruet@uhasselt.be

## Abstract

Increased soil microbial activity due to higher temperatures caused by heatwaves could lead to higher carbon losses from soil into the atmosphere, further accelerating climate change. Hence, knowing how soil microbes respond to higher temperatures is crucial for improving soil-atmosphere carbon circulation models in the context of climate change. Thermal acclimation may reduce carbon losses from soils by retarding increases in microbial activity in response to elevated temperatures. However, the capacity of soil microbes to acclimate to realistic soil temperatures in vitro, and our ability to predict this acclimation based on traits, remains poorly understood. We tested how submission to two previous temperature treatments or 'environmental histories' affected subsequent growth at two exposure temperatures for seven widespread soil saprotrophic fungi. We hypothesized that (1) soil fungi have higher intrinsic growth and weaker self-limitation after exposure to a colder environmental history, and (2) the response of fungal eco-physiological traits (pigmentation and spore load) to elevated temperature exposure predicts fungal acclimation in terms of growth. Our results showed that environmental history had limited impact on the tested fungi, and trait responses to history temperatures did not predict fungal growth after treatment with both history and exposure temperatures. We conclude that fungal thermal acclimation is limited and idiosyncratic.

## Introduction

Human-induced climate change causes irreversible ecosystem modifications and biodiversity loss, due to more frequent extreme events such as heavy rain, droughts, and heatwaves. The IPCC (Intergovernmental Panel on Climate Change) predicts a global temperature increase of 1.5°C between 2021 and 2040 [1], with heatwaves expected to intensify, occur more frequently, and have twice the impact on land surfaces by 2040 compared to 2020 [2]. The effect of climate change on ecosystem

**Data availability statement:** All relevant data are within the paper and its Supporting information files. Additionally, the full dataset has been deposited in the Zenodo repository and is available without restriction via the following link: https://doi.org/10.5281/zenodo.15010903.

**Funding:** This study was financially supported by BOF (Bijzonder Onderzoeksfonds) and FSR (Fonds Spécial de Recherche) (both Special Research Funds) from Hasselt University and University of Namur, respectively, under the research grant number BOF21DOCNA01. BOF-FSR URL: https://www.uhasselt.be/en/calls-research/detail/1195-2023-bof-joint-phd-programme-unamur-and-uhasselt The funders had no role in study design, data collection and analysis, decision to publish, or preparation of the manuscript. There was no additional external funding received for this study.

**Competing interests:** The authors have declared that no competing interests exist.

functioning will, to a large extent, depend on the activity of microbial decomposers that break down soil organic matter, as their activity leads to the release of $CO_2$ into the atmosphere [3,4]. Fungi, in particular, contribute 70–80% of microbial respiration [5,6] and 55–88% of microbial biomass [7,8].

Around half of terrestrial $CO_2$ emissions originate from microbial respiration [9], which generally increases with temperature, creating a positive feedback that accelerates climate change [10]. However, microbial acclimation to elevated temperatures can weaken this feedback by reducing the sensitivity of microbial activity to warming. Acclimation is the adjustment of physiological processes, such as growth or respiration, that enhances performance under new conditions [11–13]. In this study, the term acclimation is used to describe a specific form of phenotypic plasticity: the non-genetic, reversible adjustment of a fungus to a sustained environmental change. We chose a seven-day 'environmental history' to allow for physiological stabilization, distinguishing this response from immediate, transient thermal shocks (which occur over minutes to hours) and from long-term evolutionary adaptation (which involves genetic changes). This one-week window specifically captures the phenotypic shifts relevant to the duration of a realistic heatwave.

From a purely thermodynamical perspective, higher temperatures increase the rate of enzyme reactions up to an optimum level, beyond which enzymes denature. In contrast, cold temperatures slow reactions and reduce enzyme stability. Warm-acclimated organisms maintain enzyme stability at high temperatures, whereas cold-acclimated organisms compensate through physiological adaptations, including changes in cell membranes, and more flexible enzymes with higher catalytic efficiency than warm-acclimated enzymes [14–16].

Recent research supports the idea that thermal response in saprotrophic fungi can be complex and variable. For instance, enzyme activities and biomass production can vary across moisture and temperature gradients [17]. Also, trade-offs in fungal life history strategies were identified in a recent study, concluding that fungi with broad thermal tolerance had lower competitive ability, and that thermal niche traits had clear relationships with environmental conditions and phylogeny [18].

Furthermore, there is no consensus whether soil fungi are able to thermally acclimate, i.e., whether they show compensation to being previously exposed to lower or higher temperatures. Some studies have found that individual fungal strains reduced their activity at high temperatures after being previously exposed to a period of warm temperatures; in particular, microbial respiration was reported to be reduced in mycorrhizal fungi, and both microbial respiration and mycelial growth were shown to be reduced in saprotrophic fungi [11,19]. However, other research reported that individual fungal strains kept the ability to increase their activity along with a temperature increase, even when they were previously exposed to warmer temperatures [20,21].

The response of microbial populations to environmental change can be described using population growth [22]. Population growth entails two aspects: (1) intrinsic growth ($r_i$), which represents the per-capita growth rate of a species within a

population at low density; and (2) self-limitation ($a_{ii}$), which represents the decrease in growth rate with an increase in population size, quantifying how a species affects its own growth [23,24].

$$\frac{dN_i}{N_i dt} = r_i + a_{ii} N_i$$

Typically, the effects of environmental changes on populations are assessed by examining effects on intrinsic growth rate [25,26], while the effects on self-limitation are often ignored or assumed to be negligible. Yet, self-limitation is a crucial aspect of population dynamics, which can be affected by biotic and abiotic factors [22,27]. In the context of temperature fluctuations due to heatwaves, acclimation could manifest as changes in intrinsic growth and/ or self-limitation.

Eco-physiological traits have been found to predict population responses to changing environmental conditions [28,29]. Traits represent life history strategies that reflect evolutionary or physiological trade-offs, and their benefits depend on the environmental conditions. While the trait framework is well developed for plants, evidence suggests that microbial traits can also be used to predict environmental responses and ecosystem functions [30,31]. The recently proposed Y-A-S framework groups microorganisms into three life-strategy-based clusters: high yield (Y), resource acquisition (A), and stress tolerance (S) [32]. The microbes featuring different strategies respond differently to environmental change. Each strategy is characterized by specific traits. For example, fungal traits (hyphal length per unit litter mass loss, activity of extracellular enzymes, and the ability to grow in drought vs. reference conditions) have been examined to identify trade-offs in response to drought, and to determine the expression of Y-A-S strategies in saprotrophic fungi. They found little evidence for trade-offs in laboratory experiments, but field results suggested a trade-off between drought tolerance and enzyme activity [33,34].

Building on this work, several research gaps remain. First, there are missing empirical studies on plastic responses of microorganisms to environmental variability [12,35]. Second, there is no consensus on if and how populations of saprotrophic fungi express the capability to thermally acclimate, especially the responses of self-limitation to environmental history are unknown. Finally, it is unknown whether the acclimation capacity of fungal species can be predicted by eco-physiological traits. In this respect, two key traits are particularly interesting: pigmentation and spore load. Fungal pigmentation, a proxy of melanization, is known to be related to the stress tolerance strategy of fungal species, as it may offer protection against environmental stress [36,37]. Spore load is linked to resource discovery [38], representing resource acquisition or yield and reflecting a species' dispersal and reproductive strategy. These traits are often linked to temperature change but it is not yet clear whether the realistic temperature shifts applied in this study (18°C, 25°C) will be sufficient to cause measurable changes.

We submitted seven fungal strains to two environmental history temperature treatments (normal summer temperature or 18ºC, and elevated temperature or 25ºC), for one week within the environmental history treatment. Later, the fungi were submitted again to normal and elevated temperature within the exposure temperature treatment. We tested whether (Q1) soil fungi exhibited thermal acclimation, by measuring how the subjection to normal and elevated environmental history temperatures affected intrinsic growth and self-limitation at normal or elevated exposure temperatures. We hypothesized that individual strains would show a compensatory response to temperature, i.e., the fungal strains previously submitted to colder temperatures would have higher intrinsic growth and reduced self-limitation, compared to the samples of the same strain that previously were submitted to the elevated temperature (H1). After subjecting fungi to the environmental history treatment, we measured pigmentation and spore load to examine (Q2) whether these fungal traits could help predicting fungal capacity to acclimate, i.e., we expected these fungal traits to predict the acclimation effect on intrinsic growth and self-limitation (H2). Although soil moisture often decreases during heatwaves and influences fungal activity [17], our study focused on hot spells with temperature as a single driver. This design allows us to isolate and clearly demonstrate the role of realistic soil temperatures alone in fungal responses.

## Materials and methods

### Culturing fungal strains

We used seven fungal strains in the experiment (Table 1). They were chosen based on their wide distribution, high frequency, ecology (soil saprotrophs), and taxonomy (spread as much as possible across the fungal kingdom). All necessary collection permits were obtained via the Agentschap Natuur & Bos (ANB); permits were only required for the Mechelse Heide site. To cultivate the fungi we used Solid Modified Melin Norkrans (MMN) as growth medium. The MMN has the following composition: KH2PO4 (0.05%), NH4Cl (0.02%), MgSO4·7H2O (0.0015%), CaCl2·2H2O (0.005%), NaCl (0.0025%), FeCl3·6H2O (0.0012%), thiamine HCl (0.0001%), glucose (0.25%), and agar (1%). We chose this medium as a compromise to offer reduced nutrient levels while still supporting sufficient fungal biomass buildup. The pH of the medium was 6.5. A 9 cm diameter Petri dish was filled with 20 mL of MMN and was covered with a sterilized cellophane sheet to facilitate the collection of fungal biomass at the end of the experiment.

Before the experiment, strains were reactivated following standard laboratory protocols (S1 Text). All strains are preserved in our laboratory and are available upon request.

### Experimental design

We investigated the effects of realistic soil temperatures on intrinsic growth and self-limitation across two experimental phases, using a fully crossed factorial design (Fig 1). The experiment incorporated three factors: environmental history, exposure temperature, and strain identity.

We initially cultured each fungal strain from a 0.5 cm agar-mycelium plug and subjected them to an environmental history treatment (phase one), where they were exposed to either 18°C or 25°C for seven days, simulating the duration of a natural heatwave. Although previous studies have shown that saprotrophic fungi can acclimate over longer time periods (e.g., nine weeks; [11]), other research has reported that a seven-day incubation was enough for thermal acclimation in a few of their ectomycorrhizal fungal samples [19]. In our study, the objective was not to capture long-term acclimation, but rather to test whether fungal strains exhibit any degree of acclimation to realistic soil temperature changes in the context of a heatwave. After this period, we exposed the fungi to fresh medium at either 18°C or 25°C, marking the start of the exposure temperature treatment (phase two). This resulted in four combinations of environmental history and exposure temperature per strain for different times based on the growth rate of each strain (cf. section 'Population growth assessments') (Fig 1). The third factor, strain identity, included seven fungal strains: *Aspergillus niger*, *Cladosporium* sp., *Lycoperdon* sp., *Penicillium* sp., *Psathyrella* sp., *Trichoderma harzianum*, and *Umbelopsis* sp. (Table 1).

**Table 1. Taxonomical information on the fungal strains. Strains with the prefix 'MUCL' in the 'Origin' column came from the Belgian Coordinated Collections of Microorganisms (BCCM/MUCL, Université catholique de Louvain, Belgium) and these codes correspond to their official accession numbers.**

| Taxon | Order | Division | Origin |
|---|---|---|---|
| *Aspergillus niger* | Eurotiales | Ascomycota | Soil – Belgium (MUCL 11914) |
| *Cladosporium* sp. | Capnodiales | Ascomycota | Soil – Dry heathland in the Mechelse Heide [39] |
| *Lycoperdon* sp. | Agaricales | Basidiomycota | Soil – This research – Hasselt University in Diepenbeek Campus – Morphological identification of the sporocarp |
| *Penicillium* sp. | Eurotiales | Ascomycota | Soil – Dry heathland in the Mechelse Heide [39] |
| *Psathyrella* sp. | Agaricales | Basidiomycota | Soil – This research – Hasselt University in Diepenbeek Campus – Morphological identification of the sporocarp |
| *Trichoderma harzianum* | Hypocreales | Ascomycota | Forest -Belgium (MUCL 29707) |
| *Umbelopsis* sp. | Mucorales | Mucoromycota | Soil – Dry heathland in the Mechelse Heide [39] |

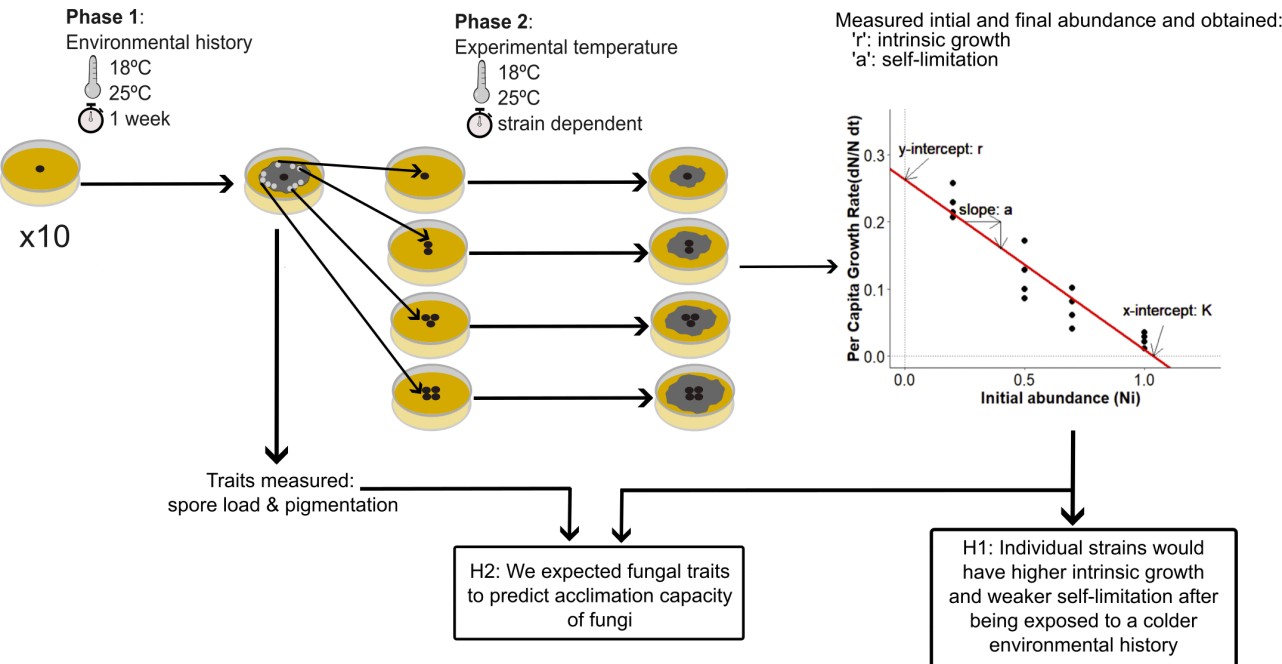

**Fig 1. Experimental design.** Per strain, during environmental history (phase one) we incubated ten petri dishes at 18ºC, and other ten petri dishes at 25ºC for one week. After that, we collected four different initial densities, to measure density-dependent growth (cf. section 'Population growth assessment'): Initial density = 0.20, 0.60, 0.77, and 1.10 cm² for *Cladosporium* sp., *Psathyrella* sp., *Trichoderma harzianum*, and *Umbelopsis* sp., and 0.08, 0.17, 0.28, and 0.34 cm² for *Aspergillus niger*, *Penicillium* sp., and *Lycoperdon* sp. We used different initial densities to create a density range in which growth was expected to respond to density. A common density range across all strains would not have been appropriate because it could have reduced our ability to estimate the density-dependence regression. For slower-growing strains, a common density range could have been too high, yielding consistently negative growth; conversely, a common density range could have been too low for faster-growing strains, yielding consistently positive growth (cf. section 'Population growth assessment'). We placed the different initial densities in new petri dishes at the exposure temperature (phase two) at 18ºC and 25ºC, in a full factorial design. After phase one we measured spore load and pigmentation, and after phase two we quantified intrinsic growth and self-limitation.

Following phase one, we measured two traits: pigmentation and spore load. Based on the data from phase two, we quantified intrinsic growth rates and self-limitation. We had ten technical replicates (Fig 1).

The temperatures represent the average soil temperatures during ambient (18°C) and heatwave (25°C) conditions during summer, in temperate-zone heathland. They are based on data from an ecotron and they reflect realistic soil temperature conditions in the first 10 cm of soil.

## Population growth assessments

The goal of phase two was to estimate density-dependent growth by regressing instantaneous per-capita growth rate against initial density. Because the instantaneous per-capita growth rate ($\frac{dN}{Ndt}$) cannot be measured directly, it was approximated as $\Delta(\ln(N))/\Delta t$, where $\Delta(\ln(N))$ is the increment of log-transformed density over a time interval $\Delta t$. Shorter intervals $\Delta t$ provide a better approximation of instantaneous growth, but if it is too short, density changes become too small to measure reliably. We therefore selected a strain-specific interval that was short enough to approximate instantaneous growth while still producing measurable biomass change. Because strains differed substantially in growth rate, slower-growing strains required longer incubation times than faster-growing strains. The growth time during phase two were as follows: *Trichoderma harzianum* (40h), *Cladosporium* sp. (45h), *Aspergillus niger* (118h), *Penicillium* sp. (122h), *Psathyrella* sp. (122h), *Umbelopsis* sp. (144h), and *Lycoperdon* sp. (186h). We acknowledge that, with this procedure, we

cannot exclude that Δt was too long, leading to a poorer approximation of the instantaneous per-capita growth rate. However, adjusting the duration by strain is expected to counteract potential bias.

Final densities of fungi were evaluated as surface, and they were calculated by taking digital scans of the petri dishes with the Epson Perfection V330 scanner. We used a calibration scale where 256 pixels were 1 cm across the whole experiment and the pixel count was converted to surface area in $cm^2$. The pixel dimensions were determined by using image analyses in Fiji [40]. First, we calculated per-capita growth rate (g) at different initial densities as

$$g = \frac{\ln(N_f) - \ln(N_i)}{\Delta t},$$

where $N_f$ was the final density, $N_i$ was the initial density and Δt was the length of the experiment.

Subsequently, we fitted a linear regression between per-capita growth rate and initial density with ten replicates per combination of initial density, strain, environmental history, and exposure temperature. Therewith, we obtained 40 data points per regression, allowing us to produce estimates for intrinsic growth (r) and self-limitation (a), represented by the intercept and slope of each regression, respectively (Fig 1, S1 Fig). By estimating intrinsic growth (r) as the y-intercept, we obtained a measure of growth at zero density ($N_i = 0$). This approach effectively decouples the intrinsic growth rate from density-dependent effects (self-limitation), providing a standardized baseline for comparison between strains.

## Surface biomass relationship

For each fungal strain, final densities were quantified as surface area data by measuring colony surface because extension rate, which is surface/time, gives a better estimate of the amount of carbon allocated than biomass data [11]. We recognize that surface area measurements on cellophane in vitro may not fully capture the complexity of fungal growth forms or biomass distribution in soil environments. Therefore, even though colony surface was used to estimate population growth, final biomass was additionally measured to later derive a more comprehensive understanding of ecosystem functions. Thus, we assessed the biomass to surface relationship of individual fungal strains at each environmental history and exposure temperature combination.

## Trait quantification

Fungal traits were measured directly after phase one (Fig 1).

We measured fungal pigmentation on the underside of the mycelium by scanning the petri dishes using an Epson Perfection V330 scanner. From each scan, we selected a $0.04\,cm^2$ area from the outermost part of the mycelium to analyze. Using Fiji GPLv3+ (version 2.15.1), we determined the Red-Green-Blue (RGB) values of this selected surface, which allowed us to assess pigmentation. To quantify the degree of pigmentation, we calculated the Euclidean distance between the RGB values for each sample ($R_{fungi}$, $G_{fungi}$, $B_{fungi}$) and the color white ($R_{white} = 255$, $G_{white} = 255$, $B_{white} = 255$), with the resulting distance representing the pigmentation level for each replicate.

$$Distance = \sqrt{\left(R_{white} - R_{fungi}\right)^2 + \left(G_{white} - G_{fungi}\right)^2 + \left(B_{white} - B_{fungi}\right)^2}$$

This formula calculates the distance between the two colors in the color space with multiple dimensions, which yields the degree of similarity between the colors. The highest value indicates the lowest degree of similarity to white and the highest degree of similarity to black; in this case, the higher the value the darker the mycelium. We used this measure of 'darkness' as a proxy for pigmentation. The background color was the same for all petri dishes in all pictures. While we

recognize that pigmentation could be more accurately quantified through biochemical extraction (i.e., assessing melanin content [41]), our digital image analysis provides an approximate but non-destructive proxy for the functional 'darkness' of the mycelium.

To calculate spores $cm^{-2}$, mycelium was carefully removed from the petri dish along with MMN agar (we took samples of a surface of $1.87\,cm^2$ for *Trichoderma harzianum*, $0.20\,cm^2$ for *Cladosporium* sp., and *Umbelopsis* sp., and $0.08\,cm^2$ for *Aspergillus niger*, and *Penicillium* sp.). These samples were transferred to Falcon tubes containing either $10\,mL$ (for *Cladosporium* sp., *Umbelopsis* sp., and *Trichoderma harzianum*) or $3\,mL$ (for *Aspergillus niger* and *Penicillium* sp.) of distilled water, and vortexed for 10 seconds to release the spores. Ten microliters of the spore solution were examined under the microscope on a Burker Haemocytometer that consists of 9 large squares of $1\,mm^2$ each. These large squares are subdivided into smaller group squares (0.2 mm sides) and those even smaller squares (0.05 mm sides). The Nikon Eclipse 80i fluorescence microscope was used at a 10x magnification to count the spores on five large squares; the average value was calculated and extrapolated to spore load per milliliter [40]. The number of spores per surface ($cm^2$) was determined by counting the average number of spores per square and applying the formula: average spores per square * dilution factor /0.0001(milliliters per each small square). The dilution factor was strain-dependent, calculated as the sampled mycelial surface area divided by the suspension volume. We provided a standardized measurement of spore production across strains. However, it is important to note that spore load was quantified at a single terminal time point. Consequently, this measurement represents a static snapshot of reproductive output and does not account for dynamic germination rates or the continuous allocation of biomass between mycelial growth and sporulation during the experiment.

## Statistical analyses

We analyzed the effects of initial densities, environmental history, exposure temperature, and the interactions between these factors on per-capita growth rate with a linear model per strain (g ~ initial_density * environmental_history * exposure_temperature). This model was used to assess how different factors influenced intrinsic growth 'r' and self-limitation 'a'. Intrinsic growth was the y-intercept of the relationship between per-capita growth rate and initial density (Fig 1). We tested its response to all factors separately, as well as their interactions, to determine how environmental history and exposure temperature influenced growth independent of initial density per strain. Self-limitation was the slope of the relationship between per-capita growth rate and density (Fig 1), representing how per-capita growth rate changes with initial density. We tested 'initial density' alone to confirm whether self-limitation was present. Then, we tested interactions of 'initial density' with all other factors to assess whether self-limitation varied across conditions. Effect sizes ($\eta^2$) were calculated to quantify the contribution of each factor to variation in intrinsic growth and self-limitation, and we used eta_squared(), which measures the proportion of variance in the dependent variable explained by an independent variable in an ANOVA. Effect size was computed as $\eta^2 = SS_{effect}/SS_{total}$, where $SS_{effect}$ is the sum of squares for a certain factor, and $SS_{total}$ is the total sum of squares in the model. Later, we performed the same linear model but across all strains (g ~ initial_abundance * environmental_history * exposure_temperature * strain) and calculated effect size for each factor.

To examine the capability of traits to predict the fungal acclimation responses, for each exposure temperature, we computed the ratios as: ratio pigmentation = pigmentation at environmental history 25ºC/ pigmentation at environmental history 18ºC, for every strain. The same was done for spore load. Finally, for each exposure temperature, we calculated a 'growth ratio' as intrinsic growth when environmental history 25ºC / intrinsic growth when environmental history 18ºC, idem for self-limitation. To assess the strength and direction of the relationship between 'ratio growth' or 'ratio self-limitation' and 'ratio pigment' or 'ratio spores', we used Spearman's rank-order correlation.

Statistical analyses were performed using R version 4.2.1 [42]. When $p < 0.05$, results were interpreted as significant. We used the packages *tidyverse* [43], *ggpubr* [44], and *effectsize* [45]. All plots were created with *ggplot2* [46].

## Results

### Effects of exposure temperature on intrinsic growth rate and self-limitation

All strains had higher intrinsic growth rate at exposure temperature 25°C, and for self-limitation the effect of exposure temperature differed depending on the strain. In particular, exposure temperature had no effect on self-limitation in basidiomycetes (*Lycoperdon* sp. and *Psathyrella* sp.) but it significantly affected most of the strains belonging to the ascomycetes and mucoromycetes (except *Cladosporium* sp. and *Trichoderma harzianum*) (Fig 2, S1 Table).

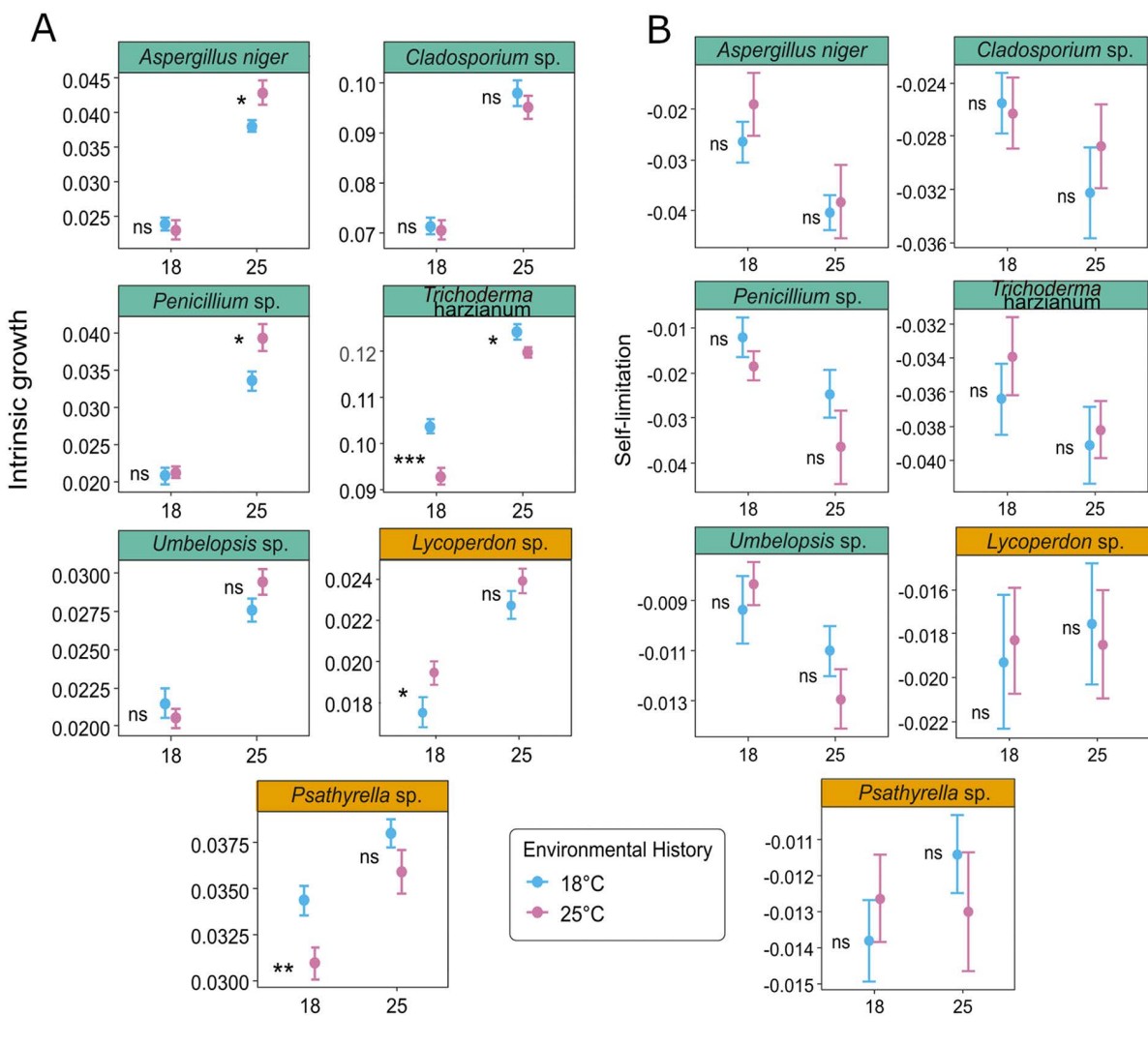

**Fig 2. Effects of environmental history and exposure temperature on population growth varied across strains: *Aspergillus niger*, *Cladosporium* sp., *Lycoperdon* sp., *Penicillium* sp., *Psathyrella* sp., *Trichoderma harzianum*, and *Umbelopsis* sp.** A. Intrinsic growth and B. self-limitation are shown for each strain, with blue and pink symbols representing environmental histories of 18°C and 25°C, respectively. The panels that belong to ascomycetes and mucoromycetes are colored in green while basidiomycetes' panels are colored in orange. Error bars indicate standard errors of the mean. Statistical significance between exposure temperatures is indicated by: ***p < 0.001, **p < 0.01, *p < 0.05, '.' p < 0.1, and 'ns' for not significant.

**Effects of environmental history on intrinsic growth rate and self-limitation**

The effect of environmental history on intrinsic growth rate was strain-dependent (Fig 2, Table 2, S1 Table), while for self-limitation, the effect of environmental history was not significant for any of the strains (Fig 2, S1 Table). *Aspergillus niger* and *Penicillium* sp. exhibited a 13% to 17% higher intrinsic growth rate when previously exposed to an environmental history of 25°C, compared to 18°C, with both species showing this increase at an exposure temperature of 25°C. In contrast, *Trichoderma harzianum* and *Psathyrella* sp. showed a 3% to 10% reduction in intrinsic growth rate under an environmental history of 25°C compared to 18°C, regardless of the exposure temperature. *Lycoperdon* sp. showed an 11% increase in intrinsic growth rate when its environmental history was 25°C compared to 18°C, only at an exposure temperature of 18°C. Finally, the intrinsic growth rates of *Cladosporium* sp. and *Umbelopsis* sp. remained unaffected by changes in environmental history across both exposure temperature treatments. In general, while environmental history and its interactions with strain and exposure temperature were statistically significant (p < 0.001), their impact was notably small within the overall variance framework. Strain identity was the dominant driver of growth dynamics, accounting for 87% of the total variance, followed by exposure temperature (6%). In contrast, the interaction between environmental history and exposure temperature accounted for less than 0.1% of the observed variance (Table 2).

We expected that fungal strains subjected to a colder temperature during phase one would exhibit higher intrinsic growth and reduced self-limitation during phase two (H1). However, our results did not support this hypothesis because the extents of these effects were strain specific (Fig 2, S1 Fig, Table 2). When considering all strains in the model, exposure temperature affected the intrinsic growth rates, which varied among strains. All the other factors and interactions significantly affected growth rate, except environmental history on its own (Table 2). Environmental history, exposure temperature, and the interaction between exposure temperature and strain affected self-limitation (Table 2).

**Table 2. ANOVA table of the linear model and effect size ($\eta^2$) of each predictor and interaction of predictors. We tested the effects of environmental history, exposure temperature, strain, and interactions on intrinsic growth and self-limitation. If values were not significant $\eta^2$ was not included. Significant values are in bold.**

| Model factors | Df | F value | p-value | $\eta^2$ |
|---|---|---|---|---|
| Initial Density | 1 | 1812.65 | **<0.001** | **0.03** |
| Environmental History | 1 | 2.59 | 0.108 | |
| Exposure Temperature | 1 | 3471.54 | **<0.001** | **0.06** |
| Strain | 6 | 8802.79 | **<0.001** | **0.87** |
| Initial Density × Environmental History | 1 | 40.18 | **<0.001** | **<0.001** |
| Initial Density × Exposure Temperature | 1 | 5.36 | 0.021 | <0.001 |
| Environmental History × Exposure Temperature | 1 | 24.13 | **<0.001** | **<0.001** |
| Initial Density × Strain | 6 | 100.97 | **<0.001** | **<0.001** |
| Environmental History × Strain | 6 | 23.55 | **<0.001** | **<0.001** |
| Exposure Temperature × Strain | 6 | 172.61 | **<0.001** | **0.02** |
| Initial Density × Environmental History × Exposure Temperature | 1 | 0.91 | 0.34 | |
| Initial Density × Environmental History × Strain | 6 | 0.63 | 0.703 | |
| Initial Density × Exposure Temperature × Strain | 6 | **2.66** | **0.014** | **<0.001** |
| Environmental History × Exposure Temperature × Strain | 6 | 4.82 | **<0.001** | **<0.001** |
| Initial Density × Environmental History × Exposure Temperature × Strain | 6 | 0.5 | 0.811 | |
| Residuals | 1016 | | | |

### Fungal surface-dry biomass relationship

Notably, fungal surface area and dry biomass were strongly associated in most strains based on the regression model (S3 Fig), highlighting a relationship between expansion and thickness. However, the biomass surface relationship varied across environmental conditions and fungal strains, as the environmental history and exposure temperature effects were strain-dependent (S3 Fig, S2 Table).

### Traits

Finally, we hypothesized (H2) that pigmentation and spore load would predict fungal acclimation, particularly in terms of intrinsic growth and self-limitation. Contrary to this expectation, our results indicated that neither pigmentation nor spore load predicted intrinsic growth or self-limitation of fungi after submission to an environmental history treatment (S4 Fig).

## Discussion

Temperature influences various biological processes, including biochemical reactions, reproduction, and development, both at the level of individual organism and at larger ecological scales, including population dynamics and ecosystem functioning [47]. The capability of soil microbes to acclimate to elevated temperature is a critically important mechanism potentially counteracting increased $CO_2$ release from soil into the atmosphere under climate warming, halting therewith the positive feedback loop of atmospheric $CO_2$ concentration. Our study is among the first to examine soil saprotrophic fungi for thermal acclimation capability (cf. [11]), and the first to address both intrinsic growth and self-limitation. Intrinsic growth rate determines a fungus's capacity to start population growth when the population is at low density. This attribute is essential for fast recovery of ecosystem functions to which fungi contribute, such as organic matter decomposition and nutrient release [48,49]. In comparison, self-limitation describes a fungus's capability to regulate its own growth and facilitate coexistence with other microbial species [50].

### Effects of environmental history and exposure temperature on intrinsic growth rate and self-limitation

While previous work [11] showed that thermal acclimation is common after a 9-week incubation, our 1-week experiment suggests that it may be more limited among heathland soil saprotrophic fungi and in the context of realistic temperature exposure. Specifically, depending on the strain, the effect of environmental history on intrinsic growth ranged from a 17% increase to a 10% decrease when comparing two temperature treatments (25°C versus 18°C).

*Umbelopsis* sp. and *Cladosporium* sp. showed no effect of environmental history on their growth rate. This is consistent with other studies where medium-term environmental history was tested, and it was observed that, among most of the species considered, fungal wood decomposers did not exhibit compensatory respiratory responses [51]. This could also be because these strains do not thermally acclimate as they gain no evolutionary advantage from physiological down-regulation as autotrophic organisms do. In plants, acclimation capability is useful because it allows keeping a positive carbon balance when they are exposed to a higher temperature, but this would not be the case for fungal decomposers [52].

We did find evidence of an effect of environmental history for *Trichoderma harzianum* and *Psathyrella* sp. Intrinsic growth was 3–10% and 5–10% lower, respectively, at the elevated environmental history at both exposure temperatures (as previously reported [11]). The fungal strains used in this study are not known to be thermophilic, and we therefore expect their optimum temperature to be between 25 and 30°C [53–55]. In this experiment, 18°C represented the normal summer soil temperature and 25°C an elevated temperature, mimicking the realistic soil temperature conditions during a heatwave. These temperatures are unlikely to induce extreme heat or cold stress, but they represent realistic conditions that fungi experience in soil. The normal soil temperature (18ºC) is below the optimum and can be considered a less favorable, or a mildly cold treatment. Thus, although temperature stress is not the main factor driving the responses observed, differences in enzyme activity can occur due to acclimation. In general, cold acclimated organisms produce

enzymes with higher catabolic rates, either by increasing the total amount of expressed enzymes, or by producing different isozymes with higher catalytic turnover, partially compensating for the effects of low temperatures [15,16,20]. Conversely, enzymes acclimated to more elevated temperatures may maintain enzyme stability and even exhibit lower activity, which is what could explain our results from *Trichoderma harzianum* and *Psathyrella* sp.

*Aspergillus niger* and *Penicillium* sp. had 13% and 17% higher intrinsic growth rates, respectively, under elevated environmental history, consistent with increases in respiration rates seen in other studies [20,21]. This could be explained because these strains may lack the genetic potential to express different isozymes, and these fungi might be more active at the closest to their optimal temperature.

Contrary to our hypothesis, environmental history did not affect self-limitation. We expected that fungi exposed to elevated environmental history treatment would show lower growth rates and higher self-limitation coefficients. It is possible that the relatively short duration of the environmental history treatment, although based on realistic heatwaves in terms of both soil temperature and length, was insufficient to show acclimation. However, our aim was not to artificially induce acclimation in the strains but to observe whether fungi naturally acclimate under realistic heatwave conditions. Moreover, our results showed that for some fungal strains self-limitation coefficients depended only on exposure temperature, suggesting that the exposure temperature directly affects metabolic and physiological processes which impact self-limitation, and these processes are not capable of acclimation.

Intrinsic growth rates were higher at 25°C exposure temperature across all strains. This is probably due to the optimization of enzymatic and metabolic processes at more elevated temperatures within physiological limits. However, self-limitation in some ascomycetes exhibited higher values at exposure temperature of 18°C, while for basidiomycetes, exposure temperature did not significantly affect self-limitation. This different response may be due to the growth conditions, as fungi were cultivated in a mineral medium (MMN) and not an organic one. However, the underlying mechanisms remain difficult to infer.

Synthesizing these responses within our broader variance framework, we determined that strain identity was the dominant driver of growth dynamics (87% of variance) followed by exposure temperature (6%). This indicates that while environmental history can induce significant physiological shifts in specific taxa, these effects are secondary to the inherent differences of the strains themselves.

## Implications for ecosystem functioning

The biomass-ratio principle states that the magnitude of a species' contribution to ecosystem function scales with its population biomass [56]. Although our assessments were based on fungal surface measurements, the tight correlation between fungal surface and fungal biomass (S3 Fig) allows us to reflect on the implications for biomass and therefore, ecosystem functioning [11,57,58]. The variability seen in the intrinsic growth rate responses to environmental history temperatures, with some fungi increasing growth while others decreasing it, would imply that ecosystem-level soil carbon turnover could shift based on fungal species composition. However, when viewed through the broader variance framework of our study, environmental history appears as a minor modulation compared to the dominant role of strain identity. Moreover, since only a minority of strains exhibited lower growth rate after being acclimated to 25°C, and when present, the effect was below 20%, the fungal acclimation phenomenon seems to have a minor, if any, impact on soil organic matter decomposition processes. Based on our in vitro results using synthetic medium, the thermal acclimation of fungal decomposers is unlikely to significantly influence ecosystem functioning, suggesting that soil saprotrophic fungi may play a limited role in mitigating carbon losses from terrestrial ecosystems or reducing the climate carbon feedback loop. While these simplified growth conditions allow for precise parameter estimation, we acknowledge that the complexity of natural soil, including recalcitrant substrates like lignin and interspecific competition, may yield different dynamics, as discussed in our limitations section.

In contrast, exposure temperature significantly affected fungal population growth. Fungi from temperate zones are known to exhibit higher activity at 25°C compared to 18°C (as 25°C is closer to their optimum temperature, as previously

noted). Our results indicate that temperature-induced changes in self-limitation observed in most ascomycetes, but not in basidiomycetes, may alter carbon cycling efficiency under changing climatic conditions. The impact of changes in self-limitation on soil processes in ecosystems would depend on the dominant fungal species in a given ecosystem.

## Traits

Our study aimed to determine whether fungal traits, specifically pigmentation and spore load, could predict the effect of environmental history of saprotrophic fungi under heatwave conditions (H2); but this hypothesis was rejected. H2 aligned with the broader Y-A-S framework, which has been used to explain trade-offs in fungal responses to environmental pressures [33]. However, our results suggest that pigmentation and spore load may not predict the effect of environmental history under realistic soil temperature change, possibly due to the complexity of fungal responses. As noted in our methods, our pigmentation proxy captured 'darkness' rather than specific melanin content; the coarse nature of our proxy might have obscured correlations. Furthermore, while spore load was standardized by surface area (spores cm$^{-2}$), this localized spore density was analyzed independently of the total colony biomass. Due to the different spatial scales and the non-uniform distribution of spores across the mycelium, a direct ratio of reproductive investment could not be reliably extrapolated for the entire individual. Consequently, we may have missed potential relationships linked to reproductive investment (i.e., the proportion of energy allocated to spores versus mycelial growth). This is particularly relevant as a high localized spore density does not necessarily reflect a higher overall investment in reproduction during the heatwave.

While both of these traits are often related to temperature [37,59], their role in thermal acclimation may be context-dependent. This complexity highlights the need for a more integrative, high-resolution trait-based approach to understanding fungal responses to environmental stressors.

## Limitations of this study and future research recommendations

Laboratory and microcosm experiments are valuable tools to disentangle key mechanisms and causal relationships in ecological processes that are challenging to examine directly in natural settings [60]. Although in our study thermal acclimation was not common among the seven soil saprotrophic fungal strains examined in isolation, these strains represent common decomposers from temperate heathland soils. Therefore, while the results cannot be directly extrapolated to entire fungal communities where diversity and interactions such as competition and predation influence dynamics, they suggest that thermal acclimation may be limited among common fungal saprotrophs in this ecosystem.

Next, our laboratory-based experimental conditions were simplified compared to the complex variability that soil fungi experience in natural environments. For instance, while self-limitation or intraspecific competition is an important component of population dynamics of an individual fungal species, in real soil communities -that consist of many fungal species- interspecific competition would also play an important role. Therefore, even if in a multispecies environment where all species coexist, self-limitation is stronger than interspecific competition [50], future studies could explore the effects of heatwaves on population parameters by including more strains to provide a more comprehensive understanding of these dynamics. Moreover, we considered only two temperatures that were close to the optima, 18 and 25 degrees Celsius, and represented ecologically relevant conditions in the context of climate change in a temperate zone. While our data showed microbial responses under these conditions, we did not account for other environmental factors, such as soil moisture. Drought also occurs during heatwaves and impacts fungal growth and activity, potentially interacting with temperature effects. Future research should include drought and examine full temperature response curves in order to obtain more fundamental insights into whether soil microbes show thermal acclimation and at what rate [61,62]. Additionally, here we only considered responses in the short term, but responses in the medium and long term may provide better predictions of microbial response to climate change [51,62]. Thus, further research is needed to obtain conclusive evidence about fungal responses in natural communities.

Furthermore, while we intuited traits like spore load and pigmentation to predict temperature effects, including other traits may improve predictive power. We did not distinguish between conidial and diffusible pigmentation, which limits the interpretation of potential functional roles. Therefore, our findings suggest that we should incorporate more traits and focus on physiological and molecular markers directly linked to thermal tolerance such as respiration rates, acidification capacity, enzymatic activities, mycelial biomass allocation, or cold/heat shock protein expression to better predict fungal growth under changing environmental conditions. This approach could increase our understanding of fungal acclimation and responses to heatwaves.

## Conclusions

We showed that seven individual species of soil saprotrophic fungi exhibited limited and strain- dependent responses of intrinsic growth rate to environmental history; and there was no effect on self-limitation. While statistically significant ($p<0.001$), the environmental history effect accounted for less than 0.1% of the total variance in our model, compared to the 87% explained by strain identity. We also demonstrated that neither pigmentation nor spore load could predict the effect of environmental history under different temperature conditions. These results indicate that under the evaluated in vitro conditions, the capacity for thermal acclimation of saprotrophic fungi to increasing heatwave temperatures appears to be limited. Therefore, further long-term field studies using complex natural substrates are needed to assess their potential role in carbon cycling in heathland ecosystems.

## Supporting information

**S1 Fig. Relationship between per-capita growth rate and initial density.** The columns represent each level of exposure temperature and the rows the different strains.
(TIF)

**S2 Fig. Top: Intrinsic growth when environmental history is 25ºC compared to the intrinsic growth when environmental history is 18ºC.** Both variables are log transformed. Bottom: Self-limitation when environmental history is 25ºC compared to the self-limitation when environmental history is 18ºC. Both plots: each strain is represented by two points: one at exposure temperature is 18ºC and another one when exposure temperature is 25ºC.
(TIF)

**S3 Fig. Relationship of final dry biomass compared to surface.** Each column is one treatment (including environmental history and exposure temperature) and each row is a strain.
(TIF)

**S4 Fig. Effect of pigmentation (upper panels) and spore load (lower panels) on intrinsic growth (top row) and self-limitation (bottom row).** Each dot (seven dots per temperature in pigmentation panels and five points in the spore panels) represents one strain, so there are two points per strain in each plot (one per temperature). Spearman's rank correlation results: growth ratio with pigmentation (18°C: $\rho = 0$, $p = 1$; 25°C: $\rho = 0.43$, $p = 0.3536$), growth ratio with spore load (18°C: $\rho = -0.2$, $p = 0.7833$; 25°C: $\rho = -0.3$, $p = 0.6833$), self-limitation with pigmentation (18°C: $\rho = -0.18$, $p = 0.7131$; 25°C: $\rho = 0.036$, $p = 0.9635$), and self-limitation with spore load (18°C: $\rho = -0.6$, $p = 0.35$; 25°C: $\rho = -0.9$, $p = 0.0833$).
(TIF)

**S1 Table. ANOVA table of the linear model per strain.** We tested effects of environmental history, exposure temperature, and interactions on intrinsic growth and self limitation per strain.
(DOCX)

**S2 Table. ANOVA table of the linear model per strain.** We tested the effects of surface, environmental history, exposure temperature, and interactions on biomass.
(DOCX)

**S1 Text. Culturing fungal strains.** Detailed description of protocols used to culture and maintain fungal strains for the experiment.
(DOCX)

## Acknowledgments

We thank Viktoriia Radchuk and Peter Kennedy for helpful discussions related to these results.

## Author contributions

**Conceptualization:** Maria Moreno-Druet, François Rineau, Nadejda A. Soudzilovskaia, Frederik De Laender.

**Data curation:** Maria Moreno-Druet.

**Formal analysis:** Maria Moreno-Druet, François Rineau, Nadejda A. Soudzilovskaia, Frederik De Laender.

**Funding acquisition:** François Rineau, Frederik De Laender.

**Investigation:** Maria Moreno-Druet, Anna Maria Kondrat, François Rineau, Nadejda A. Soudzilovskaia, Frederik De Laender.

**Methodology:** Maria Moreno-Druet, Anna Maria Kondrat, François Rineau, Nadejda A. Soudzilovskaia, Frederik De Laender.

**Project administration:** François Rineau, Nadejda A. Soudzilovskaia, Frederik De Laender.

**Supervision:** François Rineau, Nadejda A. Soudzilovskaia, Frederik De Laender.

**Writing – original draft:** Maria Moreno-Druet.

**Writing – review & editing:** Maria Moreno-Druet, Anna Maria Kondrat, François Rineau, Nadejda A. Soudzilovskaia, Frederik De Laender.

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
