## [Decision Letter · Decision Letter 0]

22 Jan 2026

PONE-D-25-53257Limited and idiosyncratic thermal acclimation in soil saprotrophic fungiPLOS One

Dear Dr. Moreno-Druet,

Thank you for submitting your manuscript to PLOS ONE. After careful consideration, we feel that it has merit but does not fully meet PLOS ONE’s publication criteria as it currently stands. Therefore, we invite you to submit a revised version of the manuscript that addresses the points raised during the review process.

We look forward to receiving your revised manuscript.

Kind regards,

Eugenio Llorens

Academic Editor

PLOS One

**Journal Requirements:**

1. When submitting your revision, we need you to address these additional requirements. Please ensure that your manuscript meets PLOS ONE's style requirements, including those for file naming. The PLOS ONE style templates can be found at https://journals.plos.org/plosone/s/file?id=wjVg/PLOSOne_formatting_sample_main_body.pdf and https://journals.plos.org/plosone/s/file?id=ba62/PLOSOne_formatting_sample_title_authors_affiliations.pdf 2. Thank you for stating in your Funding Statement: This study was financially supported by BOF (Bijzonder Onderzoeksfonds) and FSR (Fonds Spécial de Recherche) (both Special Research Funds) from Hasselt University and Namur University, respectively, under the research grant number BOF21DOCNA01BOF-FSR URL: https://www.uhasselt.be/en/calls-research/detail/1195-2023-bof-joint-phd-programme-unamur-and-uhasseltThe funders had no role in study design, data collection and analysis, decision to publish, or preparation of the manuscript  Please provide an amended statement that declares *all* the funding or sources of support (whether external or internal to your organization) received during this study, as detailed online in our guide for authors at http://journals.plos.org/plosone/s/submit-now. Please also include the statement “There was no additional external funding received for this study.” in your updated Funding Statement. Please include your amended Funding Statement within your cover letter. We will change the online submission form on your behalf. 3. In the online submission form, you indicated that your data will be submitted to a repository upon acceptance.  We strongly recommend all authors deposit their data before acceptance, as the process can be lengthy and hold up publication timelines. Please note that, though access restrictions are acceptable now, your entire minimal  dataset will need to be made freely accessible if your manuscript is accepted for publication. This policy applies to all data except where public deposition would breach compliance with the protocol approved by your research ethics board. If you are unable to adhere to our open data policy, please kindly revise your statement to explain your reasoning and we will seek the editor's input on an exemption. 4. When completing the data availability statement of the submission form, you indicated that you will make your data available on acceptance. We strongly recommend all authors decide on a data sharing plan before acceptance, as the process can be lengthy and hold up publication timelines. Please note that, though access restrictions are acceptable now, your entire data will need to be made freely accessible if your manuscript is accepted for publication. This policy applies to all data except where public deposition would breach compliance with the protocol approved by your research ethics board. If you are unable to adhere to our open data policy, please kindly revise your statement to explain your reasoning and we will seek the editor's input on an exemption. Please be assured that, once you have provided your new statement, the assessment of your exemption will not hold up the peer review process. 5. Please upload a new copy of Figure 2, S1, S2, S3 and S4, as the detail is not clear. Please follow the link for more information:  https://journals.plos.org/plosone/s/figures 6. If the reviewer comments include a recommendation to cite specific previously published works, please review and evaluate these publications to determine whether they are relevant and should be cited. There is no requirement to cite these works unless the editor has indicated otherwise.

Reviewers' comments:

Reviewer's Responses to Questions

**Comments to the Author**

1. Is the manuscript technically sound, and do the data support the conclusions?

Reviewer #1: Yes

2. Has the statistical analysis been performed appropriately and rigorously? 

Reviewer #1: Yes

3. Have the authors made all data underlying the findings in their manuscript fully available?

Reviewer #1: Yes

4. Is the manuscript presented in an intelligible fashion and written in standard English?

Reviewer #1: Yes

5. Review Comments to the Author

**Reviewer #1:** The study presents original experimental work involving seven saprobic fungal species and examines the combined effects of thermal history and thermal exposure across two experimental phases. The integration of intrinsic growth, self-limitation, pigmentation, and spore load as response variables is innovative and adds depth to the understanding of microbial thermal acclimation. This constitutes a meaningful contribution to the field.

2. Prior Publication

No issues were identified regarding prior publication or duplication of results. The dataset and analyses appear to be novel and specific to this study.

3. Experimental Design and Statistical Analyses

The experimental design is generally strong, with adequate replication (10 replicates per treatment), ecologically relevant temperatures, and appropriate use of linear regression to estimate growth parameters.

However, there are a few issues that require correction for clarity:

• In the description accompanying Figure 1 (lines 140 and 142), there is a formatting error in one of the initial density values (“0. 17” contains an extraneous space).

• The list of species associated with initial densities presents inconsistent punctuation (e.g., “Cladosporium sp., Psathyrella sp.; Trichoderma harzianum and Umbelopsis sp.”).

This may cause confusion regarding which species correspond to which density values.

Suggestion:

Please revise this section to correct numerical formatting and standardize punctuation and grouping of species.

4. Conclusions

The conclusions are well supported by the data. The findings that thermal acclimation was limited and species-dependent, that pigmentation and spore load did not predict acclimation, and that effects were generally modest (<20%) are all consistent with the analyses presented. The interpretation is balanced and does not overstate the implications.

5. Manuscript Clarity and Language

The manuscript is clearly written and the English is generally very good. Some minor improvements related to punctuation and sentence flow could help improve readability, but these are optional and do not impede understanding.

6. Ethics and Research Transparency

Although no ethical concerns are associated with the use of saprobic fungi, I recommend including explicit information on:

• The origin of the natural samples used.

• Whether any collection permits were required and obtained.

This would improve transparency, particularly for readers evaluating reproducibility.

7. Reporting Standards and Data Availability

The manuscript appears to follow appropriate reporting guidelines, and the data availability statement meets journal requirements.

Overall Assessment

This is a solid and well-executed study with clear relevance to microbial ecology and thermal biology. Only minor revisions are needed, primarily related to formatting and clarification in specific parts of the Methods section. Once these points are addressed, the manuscript will be suitable for publication.

6. PLOS authors have the option to publish the peer review history of their article (what does this mean?). If published, this will include your full peer review and any attached files.

Reviewer #1: No

---

## [Author Response · Author response to Decision Letter 1]

23 Feb 2026

We have carefully considered all comments from the Academic Editor and reviewer.

All points have been addressed in detail in the uploaded Response to Reviewers document.

Revisions include clarification of methods, correction of minor formatting issues, updated figure files (Figures 2, S1–S4), and an updated Funding Statement in the cover letter.

Figures embedded in the Word PDF may appear low-resolution; please refer to the separately uploaded high-resolution figure files.

---

## [Decision Letter · Decision Letter 1]

25 Mar 2026

PONE-D-25-53257R1Limited and idiosyncratic thermal acclimation in soil saprotrophic fungiPLOS One

Dear Dr. Moreno-Druet,

Thank you for submitting your manuscript to PLOS ONE. After careful consideration, we feel that it has merit but does not fully meet PLOS ONE’s publication criteria as it currently stands. Therefore, we invite you to submit a revised version of the manuscript that addresses the points raised during the review process.

A letter that responds to each point raised by the academic editor and reviewer(s). You should upload this letter as a separate file labeled 'Response to Reviewers'.

If you would like to make changes to y/listour financial disclosure, please include your updated statement in your cover letter. Guidelines for resubmitting your figure files are available below the reviewer comments at the end of this letter.

We look forward to receiving your revised manuscript.

Kind regards,

Eugenio Llorens

Academic Editor

PLOS One

Journal Requirements:

Reviewers' comments:

Reviewer's Responses to Questions

**Comments to the Author**

1. If the authors have adequately addressed your comments raised in a previous round of review and you feel that this manuscript is now acceptable for publication, you may indicate that here to bypass the “Comments to the Author” section, enter your conflict of interest statement in the “Confidential to Editor” section, and submit your "Accept" recommendation.

Reviewer #2: All comments have been addressed

Reviewer #3: (No Response)

2. Is the manuscript technically sound, and do the data support the conclusions?

Reviewer #2: Yes

Reviewer #3: Partly

3. Has the statistical analysis been performed appropriately and rigorously? 

Reviewer #2: Yes

Reviewer #3: Yes

4. Have the authors made all data underlying the findings in their manuscript fully available?

Reviewer #2: Yes

Reviewer #3: Yes

5. Is the manuscript presented in an intelligible fashion and written in standard English?

Reviewer #2: Yes

Reviewer #3: Yes

6. Review Comments to the Author

Reviewer #2: Figure 2: Please make the left and right panel A and B, respectively.

Methods line 149-150: Please explicitly state why initial density ranges differed by strain.

Reviewer #3: This manuscript examines whether common soil saprotrophic fungi can acclimate to realistic temperature increases associated with heatwaves, and whether key traits, such as pigmentation and spore load, can predict this capacity. In general, the authors use a well-designed experiment to address a knowledge gap in microbial feedback to climate change. However, there are issues in experimental design, data interpretation, and presentation that need to be addressed to support the conclusions.

The authors use “acclimation” as their central concept, but they fail to distinguish it from other forms of short-term phenotypic plasticity clearly. They indicate that they focus on “short-term responses (ranging from hours to days)” and therefore use the term “acclimation.” However, the experiment involves a one-week “environmental history” treatment, which does not correspond to a short-term period but rather to sustained exposure that could induce a range of responses. The authors need to define what constitutes “acclimation” in this context clearly.

The authors assess growth rates at 40-186 h to compare fast- and slow-growing strains. However, this could be misleading. For a given fast-growing strain (40 h), the measurement might capture exponential growth, whereas for a slow-growing strain (186 h), it might capture the transition to the stationary phase. The differences in the growth phases sampled could significantly affect estimates of the intrinsic growth rate and self-limitation, complicating comparisons between strains. The authors should clearly justify that the selected time points are all within the same phase of population growth for each strain. If this cannot be established, they should acknowledge it as a significant limitation of the study.

The ANOVA in Table 2 is a key strength, and its interpretation is essential. In this analysis, strain alone accounts for 87% of the variance. The next largest effect is exposure temperature (6%), and the interaction of interest (Environmental History × Exposure Temperature) accounts for <0.1% of the variance. At this point, the authors highlight that the strain-specific responses to environmental history reported in the results section (lines 277–285) show some statistically significant effects; however, the magnitude of these effects is small relative to the overall model. Accordingly, the discussion should be rephrased to emphasize that, within the overall variance framework, environmental history is a minor factor compared to species identity and current temperature.

Lines 200-227: The authors measure pigmentation as "darkness" based on an area of 0.04 cm² at the edge of the mycelium. This is an approximate method. Pigment content can be quantified biochemically. The authors should justify this method and acknowledge its limitations. Similarly, spore load is a static measurement taken after treatment. Neither spore germination nor the proportion of biomass allocated to spores versus mycelium during the growth phase is considered.

The authors claim that “the phenomenon of fungal acclimatization appears to have little, if any, impact on the decomposition processes of soil organic matter” (lines 382-386) is an excessive extrapolation. The study measured growth in a synthetic medium (MMN) under pure culture conditions. This is a far cry from the complex, competitive, and resource-limited environment of the soil. They measured growth, not the decomposition of complex organic matter such as lignin or cellulose. The authors must narrow this statement to center their conclusion on the simplified growth conditions evaluated in this in vitro study and acknowledge the need for further research under more ecologically relevant conditions.

7. PLOS authors have the option to publish the peer review history of their article (what does this mean?). If published, this will include your full peer review and any attached files.

Reviewer #2: No

Reviewer #3: No

---

## [Author Response · Author response to Decision Letter 2]

9 Apr 2026

Dear Academic Editor and Reviewers,

Thank you for your constructive and helpful feedback. We have addressed all revisions requested. Detailed responses are provided in the 'Response to Reviewers' file. Main updates include:

-Experimental design: We justified the use of strain-specific densities and incubation intervals (40-186 h) to accurately capture instantaneous growth across different fungal growth speeds.

-Methodology: We justified 'darkness' as a non-destructive proxy for pigmentation and acknowledged its limitations. We also acknowledged the limitations of static spore load measurements.

-Terminology: We clarified the definition of 'acclimation' and rephrased our conclusions to better reflect the in vitro nature of the study.

-Data interpretation: We expanded the Results and Discussion to emphasize strain-identity as a primary driver of variance.

-Figure 2: Panels A and B have been updated as requested.

We believe these changes have further strengthened the manuscript and look forward to your further consideration.

Sincerely,

Maria Moreno-Druet and the coauthors

---

## [Decision Letter · Decision Letter 2]

30 Apr 2026

Limited and idiosyncratic thermal acclimation in soil saprotrophic fungi

PONE-D-25-53257R2

Dear Dr. Moreno-Druet,

We’re pleased to inform you that your manuscript has been judged scientifically suitable for publication and will be formally accepted for publication once it meets all outstanding technical requirements.

Kind regards,

Eugenio Llorens

Academic Editor

PLOS One

Additional Editor Comments (optional):

Reviewers' comments:

Reviewer's Responses to Questions

**Comments to the Author**

1. If the authors have adequately addressed your comments raised in a previous round of review and you feel that this manuscript is now acceptable for publication, you may indicate that here to bypass the “Comments to the Author” section, enter your conflict of interest statement in the “Confidential to Editor” section, and submit your "Accept" recommendation.

Reviewer #3: All comments have been addressed

2. Is the manuscript technically sound, and do the data support the conclusions?

Reviewer #3: Yes

3. Has the statistical analysis been performed appropriately and rigorously? 

Reviewer #3: Yes

4. Have the authors made all data underlying the findings in their manuscript fully available?

Reviewer #3: Yes

5. Is the manuscript presented in an intelligible fashion and written in standard English?

Reviewer #3: Yes

6. Review Comments to the Author

Reviewer #3: The authors have adequately addressed my comments. I consider this manuscript acceptable for publication. However, they check the next confusion section before.

Lines 156-157: This section is confusing. Please revise the redaction to clarify the idea.

7. PLOS authors have the option to publish the peer review history of their article (what does this mean?). If published, this will include your full peer review and any attached files.

Reviewer #3: No

---

## [Editor Report · Acceptance letter]

PONE-D-25-53257R2

PLOS One

Dear Dr. Moreno-Druet,

I'm pleased to inform you that your manuscript has been deemed suitable for publication in PLOS One. Congratulations! Your manuscript is now being handed over to our production team.

Kind regards,

on behalf of

Dr. Eugenio Llorens

Academic Editor

PLOS One